# Coping with intimate partner violence and the COVID-19 lockdown: The perspectives of service professionals in Spain

**Carmen Vives-Cases**[1,2]*, **Daniel La Parra-Casado**[3], **Erica Briones-Vozmediano**[4,5], **Sebastià March**[6], **Ana María García-Navas**[6], **José Miguel Carrasco**[6], **Laura Otero-García**[2,7], **Belén Sanz-Barbero**[2,8]

**1** Department of Community Nursing, Preventive Medicine and Public Health and History of Science, Alicante University, Alicante, Spain, **2** Consortium for Biomedical Research in Epidemiology & Public Health (CIBERESP), Madrid, Spain, **3** Department of Sociology 2, Alicante University, Alicante, Spain, **4** Department and Faculty of Nursing and Physiotherapy, Research Group in Society, Health, Education, and Culture (GESEC), University of Lleida, Alicante, Spain, **5** Research Group in Health Care (GRECS), Biomedical Research Institute (IRB) of Lleida, Fundación Josep Pifarre, Lleida, Spain, **6** Cooperativa APLICA, Madrid, Spain, **7** Department of Nursing, Faculty of Medicine, Universidad Autónoma de Madrid, Madrid, Spain, **8** Department of Epidemiology and Biostatics, National School of Public Health, Institute of Health Carlos III, Madrid, Spain

* carmen.vives@ua.es

**Data Availability Statement:** All relevant data are within the paper.

## Abstract

Socioeconomic crisis and humanitarian disasters can cause increased stress for women who experience inter-partner violence (IPV). This study analyzed the impact of the COVID-19 lockdown on this important issue, their related health and social services and working conditions from the perspectives of professionals in different sectors. Forty-three semi-structured interviews were carried out with 47 professionals (44 women and 3 men) from 40 different entities (September 2020—April 2021). This content analysis suggests that the pandemic and its associated prevention measures have had a negative impact on women exposed to IPV and their children, which affected their social wellbeing. Professionals described burnout, difficult and slow administrative processes, and problems with coordination and access to information. These negative impacts were mitigated, in part, by the work of professionals, but this suggests that a series of key strategies are needed to improve the response capacity of the service sector to IPV in situations of crisis. These improvements are related to the availability of human and material resources; an efficient coordination network between the professionals from different sectors; existence of informal support networks in the community; protocols/procedures and prior training for better implementation; and greater flexibility and accessibility of basic services that benefit women who experience IPV.

## Introduction

The COVID-19 pandemic has intensified stress related to intimate partner violence against women (IPV), as has happened in other contexts of crisis and humanitarian disasters [1].

**Funding:** This study was financed through the project "Gender violence and social and health responses during the COVID-19 crisis" by the Fondo Supera Covid-19 CRUE-Santander for the period 2020-2021 (Ref. FSCovid19-03). It was also co-supported by the CIBER of Epidemiology and Public Health of Spain for its aid to the Gender-based Violence and Youth Research Program.

**Competing interests:** The authors have declared that no competing interests exist.

Unemployment, the temporary suspension of activity, closures of businesses and limitations in access to support resources for IPV, combined with the mandatory confinement with abusers and continued presence of minors in the home, are some of the factors commonly related to a possible worsening of cases of IPV [2].

Although many countries have reported a decrease in IPV reports and murders during the months of the COVID-19 lockdown, other indicators, such as emergency calls or surveys carried out with women and IPV service providers signal that during this period there were IPV situations that were both more severe and more frequent [3–5]. Also, new forms of abuse have occurred related to COVID-19, such as the greater occupancy of space in the household due to personal or work needs of the perpetrators; greater reproductive work of women; intentional non-compliance with protective distancing measures due to COVID-19; or, on the contrary, ridicule of women who intend to comply with them [6].

Prior qualitative studies carried out with women affected by IPV show the different situations of uncertainty and fear experienced by these women and their children during and after the COVID-19 confinement [7–9]. They also show the different difficulties found in leaving said situations of IPV, as well as the control and confrontation strategies women have developed to mitigate situations of abuse [10]. Our knowledge of the experience of IPV service providers during this period is still very limited. However, their perspectives have the potential to shed light on available information on the magnitude and intensity of cases of IPV that were not declared, the social impact of the pandemic, and above all, the different difficulties they have faced in responding to women's needs during this period [11–13].

The objective of this study was to analyze the impact of COVID-19 confinement on IPV, their related services, and working conditions from the perspective of the professionals of different sectors, including social services, health services, the police, and legal support, both on behalf of public institutions as well as the third sector in Spain. Although IPV can affect any gender, this paper is focused on IPV against women, along the lines of the framework of gender-based legislation in Spain on battered women's needs and rights [14].

## Material and methods

Qualitative study based on semi-structured interviews with professionals from different IPV-response services (from September 2020 to April 2021). The COREQ Checklist for reporting qualitative research was filled out and is attached as supporting information (S1 Table).

In order to design a heterogeneous sample, we used the following criteria: 1) type of resource (public administration; non-profit or third sector); 2) activity area (social support and care; police; legal; health); 3) area of care (emergency and detection units; integral psychosocial care and accompaniment; awareness raising units, prevention and training); 4) professional profile (technical; management; others); 5) geographic distribution by Autonomous Region; and 6) availability of specialized resources for vulnerable groups (functional diversity, Roma population). The strategy for identification, selection and recruiting of participants was carried out through an initial online prospection of potential participants, and distribution of information about the study online and the contact networks of the initially identified entities.

In total, interviews were carried out with 47 professionals (44 women and 3 men) from 40 different entities. Fifteen were services dependent on public administration bodies, 12 were community services (third sector associations with work focused on domestic violence care), and 13 health services (primary care and speciality care of the National Health System). The interviews carried out represent the regional heterogeneity of the country of Spain. Five are state level resources and the rest are resources of 13 autonomous regions. At the request of the

**Table 1. Main characteristics of the interviewed social and health services.**

| Code* | TYPE OF SERVICE/INSTITUTION | TERRITORIAL SCOPE | AUTONOMOUS REGIONS | NUMBER OF INTERVIEWS |
|---|---|---|---|---|
| GBVOS -01 | Government service | Country | - - | 18 |
| GBVOS-03 | Government service | Country | - - | |
| GBVOS-05 | Women's resource center | Local | Asturias | |
| GBVOS-06 | Government violence center | Regional | Extremadura | |
| GBVOS-08 | Office of equality | Local | Extremadura | |
| GBVOS-09 | Psychological attention center | Local | Extremadura | |
| GBVOS-12 | Office of equality | Regional | Extremadura | |
| GBVOS-13 | Women's resource center | Local | Asturias | |
| GBVOS-14 | Violence attention center | Local | Madrid | |
| GBVOS-15, 16, 17,18 | Violence attention center | Regional | Cantabria | |
| GBVOS-21 | Violence attention center | Local | Madrid | |
| GBVOS-23 | 24 hour women's center | Local | Valencia | |
| GBVOS-24 | 24 hour women's center | Local | Valencia | |
| GBVOS-26 | Psychosocial care center | Local | Madrid | |
| GBVOS-31 | Shelter for women victims of violence | Local | Valencia | |
| TS-02 | Women's association | Regional | Castilla y León | 12 |
| TS -04 | Women's association | Regional | Valencia | |
| TS -07 | Legal office | Local | Asturias | |
| TS -10 | Victims of violence association | Local | Castilla y León | |
| TS -11 | Care center for victims of violence | Regional | Catalonia | |
| TS -19 | Women's association | Country | - - | |
| TS -20 | Women's association | Local | Catalonia | |
| TS -25 | Roma women's care center | Regional | Andalucia | |
| TS -27 | Non-profit organisation | Regional | Galicia | |
| TS -28 | Women's association | Country | - - | |
| TS -35 | Women's disability association | Country | - - | |
| TS -39 | Women's association | Local | Murcia | |
| HEALTH-22 | Primary care center | Local | Madrid | 13 |
| HEALTH-29 | Hospital | Local | Aragón | |
| HEALTH-30 | Primary care center | Local | Andalucia | |
| HEALTH-32 | Primary care center | Local | Navarra | |
| HEALTH-33 | Primary care and hospital emergency room | Local | Euskadi | |
| HEALTH-34 | Hospital emergency room | Local | Madrid | |
| HEALTH-36 | Primary care center | Local | Andalucia | |
| HEALTH-37 | Primary care center | Local | Navarra | |
| HEALTH-38 | Primary care center | Local | Aragón | |
| HEALTH-40 | Hospital emergency room | Local | Andalucia | |
| HEALTH-41 | Primary care center | Local | Madrid | |
| HEALTH-42 | Hospital | Local | Valencia | |
| HEALTH-43 | Hospital | Local | Madrid | |

* Code indicates the type of resource/service: GBVOS = Gender-Based Violence Official Services; TS = Third sector; HEALTH = Health Services

participants, on four occasions two people participated together in the same interview, and on another occasion, four people with different profiles were interviewed from the same service (Table 1). The research team considered that data saturation was achieved when the latest interviews did not generate new additional information.

### Data collection

The interviews were conducted by three female researchers from the research team with training and experience in qualitative research methodology.

The interview guide (S2 Table) was based on the study objectives and was piloted and adapted in the first phases of the field work. The final structure of the guide was focused on perceptions related to: 1) context and needs of the women; 2) strategies adopted for the services and working conditions; and 3) lived work experience (obstacles, positive aspects and proposals for improvement. Interviews were conducted in Spanish and adapted to the professional profiles of the participants. Discussions within the research team were held to promote use verbal and non-verbal communication strategies to minimize possible language barriers related to the distance interviews. We carried out our interviews by telephone and video call, due to the COVID-19 context, which made physical meetings and in-person interactions difficult. The participants decided their preferred method: 22 were carried out by telephone and 21 by video call. The duration of the calls was from 50 to 70 minutes. Calls were audio recorded digitally and later transcribed. Participation was voluntary, and an information sheet was sent prior to recording, providing information about confidentiality and the policy of anonymity. Also, the participants' consent for the processing of data for research purposes was orally collected and recorded at the beginning of the interview. The study was approved by the Ethical Committee of the University of Alicante (UA-2020-07-08).

### Analysis

A qualitative content analysis was carried out using predefined tree codes that were agreed upon by the research team, based on the study objectives and a preliminary analysis of information from the interviews. After using this initial tree of codes, new codes that emerged were added during the process of analysis. The tree of codes was adapted during the coding process, by unifying codes related to similar discourses and later grouping codes to create different categories, as described in the Results section. To prevent possible bias in coding we carried out the following measures: double coding of interviews, contrasting the usefulness of emerging codes among the analysts and engaging in debate with the rest of the research team (The coding tree is shown in S3 Table). Atlas.ti (version 9) was used for this process.

## Results

Four categories showed how the professionals interviewed identified impacts related to the COVID-19 pandemic on IPV, both among women in a situation of IPV and their children as well as in the petitions for services received during the confinement period. In addition, different difficulties and facilitating factors emerged in their narratives related to the services provided in these cases during the confinement in Spain.

### Impact of the COVID-19 pandemic on women affected by IPV and their children

During the confinement, the professionals perceived a reduction in the number of deaths and reports of physical aggression. They attributed this primarily to the fact that, in being in constant contact with their aggressors, women developed strategies for mitigation through submission and conflict avoidance. They also signaled a possible increase in psychological and sexual violence, given that aggressors had more intense control of households and could commit abuse with greater impunity. They pointed out that it was not only women who were more exposed to this type of violence, it was also minors and other household members.

*An increase in intensity, actually, total isolation and an intense control over women, in the intensity and frequency of the exposure of minors who were in the household. In the end it was school that prevented them from seeing many things, wasn't it? Worse violence, and furthermore, the children seeing it. TS-19*

*The type of violence, for example, there was a resurgence in sexual aggression in the couple. . .That is to say, there were cases of women who weren't able to report it and get out of the situation. We were able to do follow-up, both psychological and social, to support them in their efforts to report it, and these sexual aggressions occurred because they were confined, because the aggressor had greater control. GBVOS-08*

Also related to the COVID-19 lockdown, the interviewees described how it may have helped women to increase their consciousness about their own IPV situation.

*It allowed them to see that their situation was tremendously unsustainable, and that they couldn't be there. Many of these women told me,' if I hadn't been confined at home with this man for so long I wouldn't have taken this step.' GBVOS-17*

*I knew that. . ., I knew she wasn't doing well, that there had been suicide intents, I knew that something was happening to her, but I didn't know why. (. . ..) Then, well, maybe this is a case in which the confinement helped bring it to light, you know? Who knows. The confinement could also have helped a bit in this way. HEALTH-30*

According to those interviewed, women who continued to live with their aggressors during confinement were more exposed to abuse (greater intensity, more frequent). This situation, linked to the aggressors' extreme supervision and control, made social interaction more difficult for these women, and therefore it was more difficult to share their fears and look for help. They felt more alone, which led them to develop strategies of conflict avoidance to contain the increase in violence.

*I think that women who are abused or who are victims of violence have tried to become invisible, in other words, tried to avoid conflict at all costs. HEALTH-42*

Those women who did not live with their aggressors also had varied experiences. The professionals interviewed described how some women could have felt safer during the confinement thanks to the lower probability of seeing their ex-partners and/or suffering abuse outside the home, which implied a reduction in their fear.

*In fact, one woman told me that she was more relaxed because she knew they were at home, that nothing could happen and that she was with her children. And, well, since they were accustomed to not going out often because of the fear they felt about going out, well, it was actually going well. GBVOS-12*

However, according to the professionals, even when they were separated, aggressors continued to try to maintain a presence in the lives of their ex-partners, by threatening them or using minors and/or food pensions as blackmail mechanisms.

*(. . .) they're already separated, they don't live with the aggressor. Then, evidently, with a lot of fear, they continue to use mechanisms of control, even without living with them. So they had*

*fear of possible reprisals, and that there would be continued abuse, attacks, that he would get the kids or show up unannounced. The level of threat continued. GBVOS-21*

*In terms of the aggressors, the truth is that the breakdowns were very few (. . .) But I'm telling you, there were many attempts, for example, insisting on seeing the kids, when they knew that they couldn't, well, it's still a way of being visible. And women had many doubts about this as well. We're in a situation, but. . .can I leave him with the child? Or not? He is insisting and saying that he's going to report me. Well, this is also real violence. GBVOS-14*

They also mentioned that, for some already separated women, the conditions of confinement and the limitations of movement made them relive the trauma of when they lived with their aggressors.

*One important thing that the women communicated to us was, for example, being shut in indoors or semi-shut in, it brought back past memories of when the aggressor would refuse to let them out of the house or shut them in. Thus, it provoked a great deal of discomfort because they had to relive these past moments. TS-27*

## Care needs during and after the COVID-19 lockdown from women exposed to IPV

Those interviewed also highlighted the women's most basic and material needs, even above the problems related to partner violence. They identified an increasing worry about feeding the children. Access to these types of resources was another of the principal petitions received by the professionals of the public administration and the non-profit sector.

*[During the pandemic] we received a ton of calls from women who couldn't pay for basic services or rent. This increased a lot, because beforehand there were few of these calls, otherwise it would have been the same. TS-04*

*At first, above all they were social, later things went back to normal daily functioning a little. But there was an increase in social demand, both in the state of alarm and later in the return to opening. GBVOS-23*

More specifically, the service professionals that offered supports related to housing also described an increase in the need for this support, especially in the first few weeks of the confinement.

*These petitions could be a little more than what I've described, the demands for rooms, right? I have to run away and need a place to stay, I need some resource that's going to cover that. As I told you, it was those first 15 days. GBVOS-21*

The interviewees also reported that during the confinement there was an increase in the demand for psychological support, primarily related to the need to speak and be listened to, and sometimes, the need for strategies for cohabitating with the aggressors. These petitions were attended to primarily by telephone.

*I didn't receive any specific petition except for women wanting to be heard, wanting support. The great majority of them were at home alone, maybe after a divorce, and because the measures that were implemented were new, they wanted to be heard. GBVOS-03*

*Now I am noticing it. During the confinement there was an extreme increase in new patients who called to get support, to vent, or because they needed psychological support. GBVOS-09*

The professionals working in the third sector during the confinement detected certain health needs, the majority of which were referred to the health system, above all, cases of anxiety and pain. Health services also reported that women in situations of IPV that had been identified specifically asked for mediation to confront the confinement living situation, and consequently they were prescribed more anti-anxiety medications.

*Well, there is more than that issue of health, of course. . . Yes, it has increased much more. Yes, there was a notable increase in health needs. I believe that now almost all of the women we are attending, they all come for health problems, like body pains, insomnia, anxiety. . . Before, there was around 60% or 70%, now it's all of them. All of the women who come to us ask to be seen by a doctor. I don't know if this has to do with the confinement, the situation of uncertainty, whether economic or social. TS-11*

*The need is, well, to give them something to support the situation. In the end that's what all women ask for when they're in such difficult situations; they ask, is there something that could help me tolerate this? All the anguish that this relationship is causing me. HEALTH-41*

## Aspects that made the response capacity of IPV services more difficult during the confinement due to COVID-19

The interviewees reported having to work in constant intensification and reorganization of activities (carrying out non-habitual work and functions), combined with insufficient personnel and infrastructure, which resulted in professionals having to lengthen their workdays, work on weekends and establish shift work in order to deliver care. Other factors added to the high workload, including the pressure to care for women's needs and the increase in individual responsibilities in the resolution of urgent cases.

*I've worked like never before in my life, and meanwhile, all of a sudden, my mother was admitted to the ICU. She was 61 years old and in 12 hours she was in critical condition. GBVOS-01*

*The social worker and the psychologist worked together to combine telework with in-person work. I went two days per week and the social worker went another two days per week, and that's how we covered four days per week. In this way, at least we were available by phone 24 hours per day, because it's true that we increased our normal workload. Our work hours usually last from 8 am to 4 pm more or less, but during the pandemic we were available, morning, afternoon and night, because the situation was exceptional for everyone. GBVOS-17*

*Nor did we have a predetermined procedure for a situation of this type. Someone would arrive all of a sudden and need help fast, and sometimes there was only one person available. It depended a lot on who was there. TS-19*

A negative aspect was the lack of experience in delivering telecare using technology. Households had to improvise to make room for work as well as reorganizing in order to attend both to work and the needs of minors and dependents that, from one day to the next, stopped being able to attend sports and educational activities, senior centres, etc. that were essential for the conciliation of work and home life.

*In two days, more or less here in the Region of Madrid, schools were closed. We are nearly 50 workers, all women, there is one man, and there was a high percentage of us caring for children. TS-19*

These factors added to the uncertainty brought about by the global pandemic and generated a negative emotional impact among professionals (frustration, powerlessness, overwhelm, fear, burnout. . .) that had a negative effect on the capacity to manage work.

*Well, first, we're very overworked. Like it or not, this also plays a role [. . .]. I feel like I'm a person who is conscientious and I try to always take that perspective, but there are times when I just don't have the energy. Times when I stay superficial and I say, 'look, there it is,' you know. No, now I don't have time or energy to take on anything more. And this obviously has an impact, because you need time to get through it. HEALTH-30*

*I'm sure that many women have had a really complicated time of it, because any patient has a hard time contacting the center. The help lines were overwhelmed, the internet visits were blocked. There were waitlists to get a call back. Sometimes so many people signed up that you couldn't call back for up to 10 days, because it was totally impossible. HEALTH-22*

In this context of adverse circumstances, another series of difficulties was mentioned in the interviews related to the care for IPV cases related to the characteristics of the resources, the barriers to women's access and the professionals themselves. In terms of the resources, the limited capacity of the public administration to adapt to the new situation is worth highlighting.

*There were difficulties in the petitions for subventions, the SEPE, all of the telematic procedures that were generated, but with difficulty and with barriers. . .The electronic DNI, the digital certificate. . .for many women there was a lack of knowledge about how to manage this. GBVOS-24*

The non-profit sector professionals reported a lack of support from institutions and the difficulties they found in contacting them. On some occasions, this was an important obstacle to attending to women's needs.

*I understand that the bureaucracy responds to the needs of the administration and so on, but we would have needed, at minimum and in the moment of maximum urgency, that it had been easier to carry out the paperwork. TS-20*

They also mentioned the problems in coordinating with other services and sectors, both those that had not been sufficiently developed prior and those in which it simply was not possible in the context of the pandemic.

*I'll give you an example from my area. The psychosocial teams from the courts have been working from home. Therefore, we could not coordinate with them. Everything has been done in the sense of, I'll leave a note with my colleague and he'll call you at home. This has made it all more difficult. GBVOS-21*

*I can tell you that we don't count on support from the administrations. In fact, what they showed us during these months is that they have been more of a hassle than a support, on various occasions. We feel pretty ignored, and of course there are basic things that we lack that they're not offering us. TS-02*

Among the barriers to access to services, they mentioned the heightened bureaucracy on which various supports depended and that were disrupted by the confinement, the weight of telework that made in-person contact difficult and the collapse of different services including social services. Many of these bureaucratic procedures switched to an online format and in the process became more complex for both the professionals and users, given the lack of training on how to use the new systems.

> *The public services were not prepared to provide a rapid and emerging response to the needs of the women. Everything required reports, a process and a protocol that we, with better coordination and response that I told you about, have improved upon. However, I still think that the administrations are a significant barrier for the women. GBVOS-08*

> *It's not that there wasn't anyone on the other side, but the bureaucracy, the level of density of the system, it didn't have the capacity that we did as grassroots organizations. In the end, in my case, if there's nothing I can do. . .well, I'll start calling and activating volunteers. A social services worker has her hands tied and is totally absorbed by the bureaucracy. TS-20*

Furthermore, they mentioned barriers to access imposed by the women's own situation of IPV, which was amplified during the pandemic.

> *There were women that were impossible to reach by any means, because they were in lockdown with their aggressor. And this is the most frustrating thing in the world. Because even when there is no confinement, the aggressor has to go to work, and there are spaces in which you can reach women. But it's that here there were many women who were impossible to reach, and there was tremendous frustration. TS-19*

According to the interviewees, the break with care services was not only a barrier to access, but it was also the loss of a source of trust between the women and the supports, or between the women and each other, which are basic tools for working on their processes. This is particularly relevant in the case of new users of the services, with whom there was no prior relationship, and also in relation to minors.

> *New users entered during the pandemic process, that is, during the state of alarm. And well, it was strange because we were accustomed to seeing them, and when you see someone in front of you, there's empathy, there's a connection. Well, it's that emotional part, right? Contact is so human, even when it's visual. Now we can't offer any type of contact, by phone. . .GBVOS-18*

> *Intervening with minors is very complicated. . .there was the issue, though it was very ordinary, that they could stay on screen. It's not the same thing for them to stay on the screen to play as it is to stay on the screen for an intervention. . .TS-10*

Mainly during the confinement, but also in the months that followed, the focus was put on a type of personalized and individualized care (psychological and emotional care and individual juridical advice) in detriment to the collective work with the women (group therapy, support groups and leisure groups, multi-topic groups: labour orientation, training in online tools), whose focus was the promotion of women's autonomy.

> *It was almost summer when we started to be able to do a little bit, for example, spaces for yoga and other activities, we activated them in the summer. And they lasted very little time, because*

*here in Catalonia we've been confined again from time to time, although it seems like now we'll be able to go out again. Imagine that this has meant reinventing projects all the time and spaces where we can carry them out, because all of sudden it was ok to do yoga, but we had to do it at another site because we needed more than two square meters, plus who knows what else, or two groups, or more hours. In the end, changes and the administration of these details has been brutal as well. TS-20*

## Aspects that facilitated the response capacity of the IPV services during the COVID-19 lockdown

Despite the difficulties already mentioned, the overall evaluation of the professionals was related to having been able to respond quickly and efficiently. Some of the factors that, in their opinions, facilitated this result were flexibility in adapting and the capacity to rapidly identify and offer solutions to the affected women, even in situations of institutional barriers and uncertainty.

*Reinventing oneself, adaptation to new situations and being there, always with the intention to collaborate for the good of the people. And above all, with lots of flexibility to adapt to new ways and to the response, demand and solutions from the perspective of accompaniment. GBVOS-17*

*Well, what we have gained in terms of rapidity and effectiveness under circumstances of extreme pressure. . . I think we knew how to provide a response to the stress. And I think we were able to adapt to using the new technologies, and I think in these times that is something really important. . .having that flexibility, knowing how to adapt, being able to provide service, even at a distance. . . TS-02*

*I also think that we reacted quickly in terms of making information available. TS-35*

They also signaled the motivation and commitment to this area of work, the availability of entities, professionals and voluntary personnel to provide care for and accompany these women in times of need. According to the interviewees, this guaranteed continuity of the care even in the most difficult times of the pandemic.

*We believe that we went above and beyond many times in our functions, as though they were our family. I think it shouldn't be that way, but well, I simply tried because of empathy, because we know these are very difficult times. . . GBVOS-12*

*There is a vocation and implication in the personnel that I think is what differentiates this from other jobs. It is what makes it different or what gives the possibility for professionals, in this type of situation, to be involved. . .and they have greater capacity for adaptation. GBVOS-15*

Another key factor according to the professionals interviewed was the development of formal support networks for coordination and collaboration with other entities and resources to provide services with greater efficiency, but also in the prior work carried out with women to support informal support networks (among the women and/or con key community agents).

*The issue of coordination was fundamental for me, and it really served us. It was a big help to be able to coordinate with social services or with all of the operators to be working together*

*towards the same goals. Even though it is true that these coordinations are habitual, perhaps in this time they were even more necessary. . . GBVOS-23*

The interviewees also reported another facilitating factor, the implementation of specific measures carried out by institutions for addressing IPV and the increase in public resources during the pandemic. These ranged from macro-level measures, such as state campaigns or specific legislation, to more concrete measures such as adaptation of tourist housing to shelter women victims of abuse who wanted to leave their homes, or the establishment of financial support. Particularly important was the recognition of those working to provide these services as essential workers.

*It is difficult, but I believe it was done well, because there was a state campaign that all estab-lishments could take in any woman that needed help. This campaign was very positive, because everyone knew about it. TS-11*

*Well look, I'm positive, it's the collective provision of a series of measures. They reinforced and bolstered those institutions that have to attend to women. For me that was the most important of all. . . We devised a series of public policies in record time to save lives, and that was really an historic achievement. GBVOS-01*

Finally, it is also worth highlighting the incorporation of new telematic tools such as What-sApp, videoconferences, and email to solve the problem of accessibility during the confine-ment, which are considered relevant for continuing to maintain.

*You could provide services during the first part by telephone, or by videoconference, through WhatsApp or email, you know? There were many possibilities to get access if you wanted it. And they were easy. GBVOS-09*

*We are already offering, in our next projects, the option to do a videoconference. We have to determine how, when and in which cases. TS-19*

## Discussion

From the perspective of the professionals interviewed, the COVID-19 pandemic and the related health measures have had a negative impact on the health and wellbeing of women affected by IPV. In their opinions, the confinement generated an increase in tension and con-trol of women who live with their partners and abusers. During the confinement, women demanded psychological support, accompaniment and to be heard, in an important way. They also reclaimed other needs related to the basic needs of themselves and their children, which shows their situation of extreme social vulnerability.

The professionals interviewed reported a lack of resources to respond to the needs of women during the pandemic, and they described how services became saturated with increased demand due to the pandemic. Some of the professionals also identified the lack of prior training needed to attend to the complexity of IPV in an emergency context such as that of COVID-19. Despite these difficulties, many had a positive evaluation of their actions and highlighted the level of professional commitment, the use of measures and tools adapted to the context of the pandemic, and the consideration of their services as essential [15].

The perceptions of the professionals regarding the negative impact of the COVID-19 pre-vention measures on the severity and frequency of IPV cases coincides with what has been reported by IPV victims in other studies [7–9]. Although in some countries [4, 5] including

Spain [16] this negative impact did not correspond to an increase in reporting and deaths due to IPV, the testimonials of the professionals show the damaging consequences of social isolation and uncertainty generated by the prevention measures as well as the socioeconomic consequences for the population. Especially noteworthy are the special risks of women who had to live with their aggressors during the COVID-19 confinement [2, 17–20].

This study showed the impact on women that did not live with their aggressors, a situation with little coverage in the literature [1, 6]. Our results confirm the persistence of other types of violence exercised by ex-partners who no longer live with survivors, such as threats directed towards women and other types of coercion that involve children. They also show how the COVID-19 control measures, such as forced isolation or even the use of masks in the street [21] can contribute to women reliving past trauma.

Among petitions received by the professionals, those related to basic needs are worth highlighting. These included paying the bills, the need for food or access to a secure living space, which have also been reported in other countries [5, 11, 13, 22]. The pandemic has generated a generalized socioeconomic crisis that, in the case of women who experience IPV, has compromised their possibilities to break with a relationship of violence. It has also resulted in influencing their risks (new cases) and the severity of IPV [17, 23]. It is worrisome that even after months of COVID-19 confinement, these consequences continue for these women, their children and the risk of IPV.

Also among the demands of the women was the call for psychological help for mental health problems related not only to IPV exposure, but also the generalized uncertainty and its consequences, for example, in terms of unemployment [9]. During the COVID-19 confinement, in-person primary care was limited [24]; primary care is generally the port of access to other specialties such as mental health. At the same time, other services such as telephone emergency services (016) expanded such that a special WhatsApp line was put into place [25]. The 016 line in Spain is a specialized service that supports victims of IPV by providing information and legal support as well as psychological support [26]. The increase of more than 45 percent, compared to 2019, in 016 calls during the pandemic months of COVID-19 confinement in Spain [16] suggests the magnitude of the needs of affected women for these types of services.

Problems of coordination, workload, scarcity of resources, lack of training and limited flexibility of the services mentioned by the professionals coincide with what has been reported in other contexts [11] and at other times during the months of COVID-19 confinement [27–32]. The tension produced by the social and health emergency, above all during the first months, has increased the tension of professionals due to the limitations in available resources was added to the increased demands for care, access barriers for women due to the measures in palace and the need to adopt protection measures for COVID-19 [33, 34].

In other contexts of crisis, it has been observed that the implication and motivation of professionals tends to be one of the primary factors that helps them manage barriers and difficulties that exist in fulfilling their professional competencies [35]. However, this personal commitment alluded to by the professionals requires both short and long-term improvements in the tools and conditions of the context of their work [29].

It is interesting to note the differences in care in the three sectors of professionals interviewed. The health sector completely changed its care in the pandemic, leaving aside aspects such as the detection of IPV in women. Some services of the administration, of general care but fundamental for the attention to IPV, such as social or judicial services, were paralyzed or overwhelmed during the pandemic. On the other hand, the specific services interviewed for IPV care, of the administration and the third sector, continued working, although with difficulties. The services of the third sector were perceived as more adaptable than those of the state administrations. Precisely the lack of adaptability of public services that added to the

rigidity of its bureaucracy and the closure of some of its general care services. This posed important barriers for IPV care. For future emergency situations, continuity of care should be maintained for these women, access and procedures for processing aid and resources should be facilitated, and there should be better coordination between services, in order to reduce the barriers women face to getting IPV care.

In terms of limitations, it is worth noting that those interviewed were only able to provide a description of events based on the information provided to them by those they had contact with. They were not able to provide information about emerging situations of violence that they did not witness in their workplaces. However, their perceptions are along the lines of what has been observed in other studies of the perceptions of women. In addition, it should also be considered that the interviews were carried out between three and six months after the months of COVID-19 confinement. It could be that their perceptions of the problems and facilitators in IPV care were influenced by the post-confinement context. In this study, we aimed to collect the testimonies of a heterogeneous sample that could be considered representative of the diversity of formal IPV services. However, it is necessary to continue to go deeper into the concrete experiences of professionals who came into direct contact with women in situations of social vulnerability due to disability, migratory status, belonging to sexual or ethnic minorities, young women and adolescents or elderly women. Also, we were unable to record the experience of professionals from social and judicial services agencies who play an important role in IPV care. Additional communication with the professionals from the IPV-response services to receive feedback of the findings was not carried out, because of the high workload experienced by the groups during this period.

The study also includes several strengths related to the application of an emerging design, a theoretical sample based on the quality of the information obtained and based on open seeking to record the experience of the diverse entities involved in IPV response in different phases of the field work, the level of final saturation achieved, the provision of a description both the context of the interviewees and the use of triangulation through selecting participants with different profiles and a combination of individual and two-persons interviews. Furthermore, there is novelty in exploring an approximation to the impact of the COVID-19 prevention measures on IPV from the perspectives and experiences of the professionals that worked on the front lines in attending to the needs of women during the months of confinement.

## Conclusions

From the perspective of the first line care providers of different services, the confinement and other health measures related to COVID-19 in Spain has had a negative impact on women affected by IPV and their children. This impact concerns not only the frequency, severity and suffering generated by the women's (both those living with and without their aggressors) own IPV situations, it also affects women's social wellbeing and that of their children. Professionals reported having experienced situations of burnout, difficult and slow administrative processes and problems with coordination and access to the information needed to properly respond to the needs of women and their children. These negative impacts were in part mitigated by the involvement of the professionals.

A combined reading of these problems as well as the facilitators mentioned supports identifying a series of key strategies to improve the response capacity of health and social services in times of crisis [13]. These have also been described in previously published literature concerning health services in general [36]: availability of human resources and materials needed, adapted to the needs of the occasion; a coordination network that includes professionals from different public sectors and the non-profit sector; existence of informal support networks in

the community; protocols and procedures as well as prior training to improve implementation; simplification of processes and greater flexibility and accessibility of basic services in support of women and the professionals themselves.

## Supporting information

**S1 Table. COREQ checklist.**
(DOCX)

**S2 Table. Interview guide questions.**
(DOCX)

**S3 Table. Coding tree.**
(DOCX)

## Acknowledgments

To all the study participants for their generous contributions and their high professionalism in attending to the daily needs of women in situations of IPV. To the Serra-Hunter University Program of the Generalitat de Cataluña for the support to one of the authors (EBV).

## Author Contributions

**Conceptualization:** Carmen Vives-Cases, Daniel La Parra-Casado, Belén Sanz-Barbero.

**Data curation:** Sebastià March, Ana María García-Navas, José Miguel Carrasco.

**Formal analysis:** Carmen Vives-Cases, Sebastià March, Ana María García-Navas, José Miguel Carrasco.

**Funding acquisition:** Carmen Vives-Cases, Laura Otero-García.

**Investigation:** Carmen Vives-Cases, Daniel La Parra-Casado, Erica Briones-Vozmediano, Laura Otero-García, Belén Sanz-Barbero.

**Methodology:** Carmen Vives-Cases, Daniel La Parra-Casado, Erica Briones-Vozmediano, José Miguel Carrasco, Laura Otero-García, Belén Sanz-Barbero.

**Project administration:** Carmen Vives-Cases.

**Supervision:** Carmen Vives-Cases, Belén Sanz-Barbero.

**Validation:** Erica Briones-Vozmediano.

**Visualization:** Erica Briones-Vozmediano.

**Writing – original draft:** Carmen Vives-Cases.

**Writing – review & editing:** Carmen Vives-Cases, Daniel La Parra-Casado, Erica Briones-Vozmediano, Sebastià March, Ana María García-Navas, José Miguel Carrasco, Laura Otero-García, Belén Sanz-Barbero.

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
