## [Decision Letter · Decision Letter 0]

9 Aug 2021

PONE-D-21-22176

Coping with Intimate Partner Violence and the COVID-19 lockdown: The Perspectives of Service Professionals in Spain

PLOS ONE

Dear Dr. Vives-Cases,

Thank you for submitting your manuscript to PLOS ONE. After careful consideration, we feel that it has merit but does not fully meet PLOS ONE’s publication criteria as it currently stands. Therefore, we invite you to submit a revised version of the manuscript that addresses the points raised during the review process.

We look forward to receiving your revised manuscript.

Kind regards,

Alok Atreya

Academic Editor

PLOS ONE

Journal Requirements:

2. Please include additional information regarding the interview guide used in the study and ensure that you have provided sufficient details that others could replicate the analyses. For instance, if you developed a interview guide as part of this study and it is not under a copyright license more restrictive than CC-BY, please include a copy, in both the original language and English, as Supporting Information.

Reviewers' comments:

Reviewer's Responses to Questions

**Comments to the Author**

1. Is the manuscript technically sound, and do the data support the conclusions?

Reviewer #1: Yes

Reviewer #2: Yes

2. Has the statistical analysis been performed appropriately and rigorously? 

Reviewer #1: N/A

Reviewer #2: Yes

3. Have the authors made all data underlying the findings in their manuscript fully available?

Reviewer #1: No

Reviewer #2: Yes

4. Is the manuscript presented in an intelligible fashion and written in standard English?

Reviewer #1: No

Reviewer #2: Yes

5. Review Comments to the Author

Reviewer #1: Review report

This qualitative study has explored the situation of intimate partner violence (IPV) in relation to COVID -19 lockdown in Spain from the perspective of service providers responding to IPV. The authors collected data through interviews using telephone and video calls with 47 service providers from various sectors namely non-profit social support organizations, public administrative, legal and health care services situated in various autonomous areas of Spain. They conducted qualitative content analysis of the interview transcripts and obtained their findings in terms of the perception and experiences of the service providers. The authors report their findings under four categories/themes: (1) the impact of COVID-19 pandemic and its preventive measures on IPV (frequency, intensity, types or forms), (2) care needs of those affected by IPV, (3) challenges faced by the service providers and (4) factors that facilitated the response capacity of the service providers. The authors conclude that the pandemic and lockdown, as a whole, made the IPV situation worse and they illustrate the additional psychological and social adversities faced by the affected women. They highlight the increased care needs of women affected by IPV and their children with various examples. The study also suggest some strategies to address it that include efficient coordination among various stakeholders and capacity building.

Overall, I appreciate the study and believe that its contribution to the literature is valuable. Further, I suggest that the following comments be considered before the study is published.

Major comments:

1. The authors need to provide further details of the data collection process in the methods section. Consolidated criteria for reporting qualitative studies (COREQ) may provide guidance in this regard. Details about the interviewer/s and the levels of responsibility/expertise of the participants in their respective fields are lacking. Information about the language/s used in the interviews including measures adopted to prevent possible language barriers are missing. The authors state (second paragraph under ‘data collection’ subheading, line 107) that they carried out the interviews by ‘telephone and video call’ due to the COVID-19 context. However, it is not clear which one was the major medium used and why it was not made uniform. Also, it is desirable to add a note if the authors tried to minimize the shortcomings of these distance interviews.

2. Methods and materials section, second paragraph, lines 109-111: “Participation was voluntary and with informed consent prior to recording, which guaranteed confidentiality and anonymity of the participants.” I find this statement unclear and I suggest that it is clarified. The form of consent (oral or written) also needs to be mentioned.

Besides this, it is necessary to make sure that the participants did agree to have their quotations published.

3. Methods and materials section, ‘analysis’ subheading: Description of coding tree is lacking. Measures taken, if any, to prevent bias in coding might be discussed. If the authors did communicate their findings with the participants for their feedback/verification, it is better discussed. Or, these limitations might be acknowledged.

4. It is possible that intimate partner violence can affect any gender. This manuscript seems to focus on IPV against women only. Was it the aim of the study from the beginning? If so, it is desirable to be reflected explicitly, including in the title and the abstract. Or, the findings related to IPV against men also need to be presented and discussed.

5. What were the similarities and the differences in the perceptions of the participants from different sectors namely social support organizations, public administration and health care? Were there any contradictory perceptions? Explicit discussion of these aspects might make the analysis more easily comprehensible.

Other comments:

1. In the last paragraph of the discussion section (line 484), use of ‘theoretical sample based on the quality of the information obtained’ has been stated as a strength of the study. It has not been discussed clearly how it applies to this study. Also, the concept of level of full saturation has been mentioned (lines 484, 485) as a strength of the study but not discussed clearly in the methods section.

2. The manuscript including the abstract needs some editing for following purposes.

a. For grammatical and punctuation correction (e.g. on using quotations within quotations in the results section, also in the references section), consistency in style (formats of subheadings under results section)

b. For clarity in expression. Examples:

Lines 29-31: To make the message understandable without referring to the full manuscript, this sentence might be split and examples of the response services might be added.

More suitable alternatives to the following words/phrases might express the meanings better in line 35 (….referred to…), line 37 (…involvement of professionals….), 38 (….require….), 87 (…dissemination of the study…), 109 (…literally transcribed…), etc.

Lines 98, 122-124, ………., 428-430, 460-464: Contain phrases/statements that might be rephrased for clarity.

3. Data collection (line 102): The authors refer to ‘Annex 1’ for the interview guide but I could not access it. I am not sure to what extent data availability applies to this study.

Finally, I have noted the following strengths of the study.

1. It explores an important issue and contributes significantly to the literature.

2. This study suggests that the decrease in the number of officially reported cases of IPV by other data should be interpreted cautiously. In this regard, the present study supports an alternative explanation based on the perception of the front-line service providers.

3. It also broadens the understanding of the newer forms of IPVs in the COVID -19 context.

4. Furthermore, it suggests potential service strategies from the view-points of different service providers thus representing a broad overview. It can be useful to the stakeholders while planning or executing service delivery and formulating new policies.

5. Qualitative approach can be expected to have explored the perception and experience of the service providers in a considerable depth.

(The end)

Reviewer #2: The manuscript has been prepared well. All the sections of the article has been well written. The results are presented as per the objectives and has been well analysed.

However, there few areas where some additions could be done.

In the abstract, there is no mention of the background of the study and it has straight away started with the study objectives.

Likewise, it has no mention of bias in the sampling done.

Strengths of the study has been mentioned but there is no mention of limitations of the study in the discussion section.

6. PLOS authors have the option to publish the peer review history of their article (what does this mean?). If published, this will include your full peer review and any attached files.

Reviewer #1: No

Reviewer #2: No

---

## [Author Response · Author response to Decision Letter 0]

24 Sep 2021

Reviewer #1: Review report

This qualitative study has explored the situation of intimate partner violence (IPV) in relation to COVID -19 lockdown in Spain from the perspective of service providers responding to IPV. The authors collected data through interviews using telephone and video calls with 47 service providers from various sectors namely non-profit social support organizations, public administrative, legal and health care services situated in various autonomous areas of Spain. They conducted qualitative content analysis of the interview transcripts and obtained their findings in terms of the perception and experiences of the service providers. The authors report their findings under four categories/themes: (1) the impact of COVID-19 pandemic and its preventive measures on IPV (frequency, intensity, types or forms), (2) care needs of those affected by IPV, (3) challenges faced by the service providers and (4) factors that facilitated the response capacity of the service providers. The authors conclude that the pandemic and lockdown, as a whole, made the IPV situation worse and they illustrate the additional psychological and social adversities faced by the affected women. They highlight the increased care needs of women affected by IPV and their children with various examples. The study also suggest some strategies to address it that include efficient coordination among various stakeholders and capacity building.

Overall, I appreciate the study and believe that its contribution to the literature is valuable. Further, I suggest that the following comments be considered before the study is published. 

Major comments:

1. The authors need to provide further details of the data collection process in the methods section. Consolidated criteria for reporting qualitative studies (COREQ) may provide guidance in this regard. Details about the interviewer/s and the levels of responsibility/expertise of the participants in their respective fields are lacking. Information about the language/s used in the interviews including measures adopted to prevent possible language barriers are missing. The authors state (second paragraph under ‘data collection’ subheading, line 107) that they carried out the interviews by ‘telephone and video call’ due to the COVID-19 context. However, it is not clear which one was the major medium used and why it was not made uniform. Also, it is desirable to add a note if the authors tried to minimize the shortcomings of these distance interviews.

Following the reviewer's recommendation, the COREQ checklist has been filled in and was added as a complementary annex (See Annex 1). We added a line of text in the method section to link it (see page 4): 

The COREQ Checklist for reporting qualitative research was filled out and attached as a Supplementary Annex.

Regarding the characteristics of the interviewer/s, the interviews were carried out by three different interviewers from the research team, all of whom had extensive training and experience in qualitative methodology and in conducting interviews. Although all three were women, gender was not a criteria in selecting the interviewers. The professional profiles of the participants was considered. The main content of the interviews was related to their role as professionals. Information on these aspects was added in the “data collection” subsection of the methods section (see page 6):

The interviews were conducted by three female researchers from the research team with training and experience in qualitative research methodology. 

The language used to carry out the interviews was Spanish because participants are Spanish speakers, as is the research team. Use of Spanish facilitated communication during the interview and allowed for a more accurate understanding of the meaning of what was said. No language or cultural barriers were detected. The research team conducted opening discussions during different sessions to prevent barriers and facilitate communication in the interviews, their conclusions were: 1) to adapt the language to the professional profiles of the participants 2) to use non-verbal communication strategies to minimize the possible shortcomings of these distance interviews. 

We added this sentence to clarify this aspect in “data collection” subheading (see page 7):

Interviews were conducted in Spanish and adapted to the professional profiles of the participants. Discussions within the research team were held to promote use verbal and non-verbal communication strategies to minimize possible language barriers related to the distance interviews.

The type of interview was either a telephone or video call, depending on the preference of the interviewees. Video and telephone calls were chosen equally, there was a distribution of about half and half. Despite that during COVID-19 video has become one of the main means of communication, some people did not feel comfortable using it. This information was added in the “data collection” subheading (see page 6): 

We carried out our interviews by telephone and video call, due to the COVID-19 context, which made physical meetings and in-person interactions difficult. The participants decided their preferred method: 22 were carried out by telephone and 21 by video call.

2. Methods and materials section, second paragraph, lines 109-111: “Participation was voluntary and with informed consent prior to recording, which guaranteed confidentiality and anonymity of the participants.” I find this statement unclear and I suggest that it is clarified. The form of consent (oral or written) also needs to be mentioned. Besides this, it is necessary to make sure that the participants did agree to have their quotations published.

Thank you for the recommendation. We have included in the manuscript a reference to the process we followed to inform the interviewees about confidentiality and the policy of anonymity in the treatment of data. Data was collected strictly for the purposes of the objectives of the study. The process consisted of sending an information sheet to each interviewee that described these aspects. At the beginning of the interviews, the participants had the opportunity to ask questions about the study and to share concerns. In addition, an oral informed consent was collected and recoded prior to the start the interview. This additional information was added, described on page 7 of the methods section: 

The duration of the calls was from 50 to 70 minutes. Calls were audio recorded digitally and later literally transcribed. Participation was voluntary, and an information sheet was sent prior to recording, providing information about confidentiality and the policy of anonymity. Also, the participants’ consent for the processing of data for research purposes was orally collected and recorded at the beginning of the interview. The study was approved by the Ethical Committee of the University of Alicante (UA-2020-07-08).

3. Methods and materials section, ‘analysis’ subheading: Description of coding tree is lacking. Measures taken, if any, to prevent bias in coding might be discussed. If the authors did communicate their findings with the participants for their feedback/verification, it is better discussed. Or, these limitations might be acknowledged.

Following your suggestion, we have added the coding tree as a supplementary annex (see Annex 3). Three researchers worked together to draw up the coding tree. They carried out double coding of interviews and contrasted the usefulness of the emerging codes between the analysts, discussing doubts and reflection with the rest of the members of the research team. These measures were taken to prevent possible bias in coding. 

After use of the initial coding tree, new codes that emerged were added during the analysis process. The tree of codes was adapted by unifying codes related to similar discourses and later grouping codes to create different categories, as described in the results section. To prevent possible bias in coding we carried out the following measures: double coding of interviews, contrasting the usefulness of emerging codes among the analysts and engaging in debate with the rest of the research team. The coding tree has been added as a supplementary annex (See these details on page 7 of the current version of the manuscript). 

Because of the high workload of the professionals at IPV-response services, we decide against recontacting them. Thus, their feedback was not collected. The period following the confinement was characterized by an overload of work for the workers of these services. A reference to this aspect was added in the “discussion” section as a limitation of the research (see page 22):

Additional communication with the professionals from the IPV-response services to receive feedback of the findings was not carried out, because of the high workload experienced by the groups during this period.

4. It is possible that intimate partner violence can affect any gender. This manuscript seems to focus on IPV against women only. Was it the aim of the study from the beginning? If so, it is desirable to be reflected explicitly, including in the title and the abstract. Or, the findings related to IPV against men also need to be presented and discussed.

Although IPV can affect any gender, this paper is focused on IPV against women, along the lines of the framework of gender-based legislation in Spain on battered women’s needs and rights. We agree that intimate partner violence changed for both genders, but considering men was beyond the scope of this study. To be clear about this, we added information in the abstract and the introduction, as suggested. We did not include this information in the article's title because of the total number of words. 

5. What were the similarities and the differences in the perceptions of the participants from different sectors namely social support organizations, public administration and health care? Were there any contradictory perceptions? Explicit discussion of these aspects might make the analysis more easily comprehensible.

In the manuscript, we tried to reflect the similarities and differences in the perceptions of the participants from different sectors. In general, there were few differences in the perceptions of the people attended: there was a negative impact of the COVID-19 pandemic on women affected by IPV and their children (greater exposure to violence, intensification of psychological effects, etc.), material and care needs during and after the COVID-19 lockdown among women exposed to IPV. The demands were specific of each resource, but we tried to identify a common discourse by identifying similarities. The great differences were largely related to the response of each of these different sectors (public administration, third sector and health care) to the crisis. They developed different strategies and showed different adaptability. Some references show how professionals from one sector (third sector) hold a critical stance regarding the response of public administration bodies. This information can be found on page 14 in the “results” section: 

In terms of the resources, the limited capacity of the public administration to adapt to the new situation is worth highlighting. The non-profit sector professionals reported a lack of support from institutions and difficulties in contacting them. On some occasions, this was an important obstacle to attending to women’s needs.

In the results section of the manuscript, we tried to indicate instances in which the discourse belonged to one of those collectives or to all of them. Also, following the advice of the reviewer, we have added the following paragraph in the discussion (see page 22):

It is interesting to note the differences in care in the three sectors of professionals interviewed. The health sector completely changed its care in the pandemic, leaving aside aspects such as the detection of IPV in women. Some services of the administration, of general care but fundamental for the attention to IPV, such as social or judicial services, were paralyzed or overwhelmed during the pandemic. On the other hand, the specific services interviewed for IPV care, of the administration and the third sector, continued working, although with difficulties. The services of the third sector were perceived as more adaptable than those of the state administrations. Precisely the lack of adaptability of public services that added to the rigidity of its bureaucracy and the closure of some of its general care services. This posed important barriers for IPV care. For future emergency situations, continuity of care should be maintained for these women, access and procedures for processing aid and resources should be facilitated, and there should be better coordination between services, in order to reduce the barriers women face to getting IPV care.

Other comments:

In the last paragraph of the discussion section (line 484), use of ‘theoretical sample based on the quality of the information obtained’ has been stated as a strength of the study. It has not been discussed clearly how it applies to this study. Also, the concept of level of full saturation has been mentioned (lines 484, 485) as a strength of the study but not discussed clearly in the methods section.

The aim of using a theoretical sample design is to provide the ability to collect data from a diverse group of entities involved in the response to IPV. We obtained a sufficiently heterogeneous sample of agents who participated in the IPV response. Following the reviewer’s advice, we made modifications in the “discussion” section to clarify this (see page 23):

Theoretical sample based on the quality of the information obtained and based on open seeking to record the experience of the diverse entities involved in IPV response in different phases of the field work. 

However, it would have been interesting to record the experience of professionals who work in the field of social and judicial services, because they play an important role in IPV care. Unfortunately, we could not reach them at the moment of the study. A reference to this limitation of the study was added in the “discussion” section (see page 23):

Also, we were unable to record the experience of professionals from social and judicial services agencies who play an important role in IPV care. 

A reference to the level of saturation reached has been included in the manuscript. Thematic saturation was reached in the last interviews carried out: the last interviews did not provide new data or additional information. Information on this aspect can be found in the “material and methods” section (see page 5): 

The research team considered that data saturation was achieved when the latest interviews did not generate new additional information.

2. The manuscript including the abstract needs some editing for following purposes.

a. For grammatical and punctuation correction (e.g. on using quotations within quotations in the results section, also in the references section), consistency in style (formats of subheadings under results section) 

We have rephrased the referenced sections and corrected punctuation where needed. 

b. For clarity in expression.

 Examples:

Lines 29-31: To make the message understandable without referring to the full manuscript, this sentence might be split and examples of the response services might be added.

These lines have been rewritten. 

More suitable alternatives to the following words/phrases might express the meanings better in line 35 (….referred to…), line 37 (…involvement of professionals….), 38 (….require….), 87 (…dissemination of the study…), 109 (…literally transcribed…), etc.

We have found suitable alternatives to the words and phrases requested. 

Lines 98, 122-124, ………., 428-430, 460-464: Contain phrases/statements that might be rephrased for clarity.

We have rephrased the lines mentioned for greater clarity. 

3. Data collection (line 102): The authors refer to ‘Annex 1’ for the interview guide but I could not access it. I am not sure to what extent data availability applies to this study.

This has been added again (now as Annex 2).

Finally, I have noted the following strengths of the study.

1. It explores an important issue and contributes significantly to the literature.

2. This study suggests that the decrease in the number of officially reported cases of IPV by other data should be interpreted cautiously. In this regard, the present study supports an alternative explanation based on the perception of the front-line service providers.

3. It also broadens the understanding of the newer forms of IPVs in the COVID -19 context.

4. Furthermore, it suggests potential service strategies from the view-points of different service providers thus representing a broad overview. It can be useful to the stakeholders while planning or executing service delivery and formulating new policies.

5. Qualitative approach can be expected to have explored the perception and experience of the service providers in a considerable depth.

We are very grateful for these comments. 

Reviewer #2: The manuscript has been prepared well. All the sections of the article has been well written. The results are presented as per the objectives and has been well analysed.

We appreciate these positive comments. 

However, there few areas where some additions could be done.

In the abstract, there is no mention of the background of the study and it has straight away started with the study objectives.

We have added the following in the new version of the manuscript (see page 2): 

Socioeconomic crisis and humanitarian disasters have been described as the cause of intensified stress related to intimate partner violence against women (IPV). In this paper we analyzed the impact of COVID-19… 

Likewise, it has no mention of bias in the sampling done.

The sample selection has been explained in more detail (see page 5), and it has been mentioned as a possible source of bias in the discussion section (see pages 21-22). 

Strengths of the study has been mentioned but there is no mention of limitations of the study in the discussion section.

We have expanded the section concerning limitations with two new paragraphs, in line with the comments made by both reviewers (see pages 21 and 22):

It is interesting to highlight the differences in care in the three sectors of professionals interviewed. The health sector completely changed its care in the pandemic, leaving aside aspects such as the detection of IPV in women. Some services of the administration, of general care but fundamental for the attention to IPV, such as social or judicial services, were paralyzed or overwhelmed during the pandemic. On the other hand, the specific services interviewed for IPV care, of the administration and the third sector, continued working, although with difficulties. The services of the third sector were perceived as more adaptable than those of the state administrations. Precisely the lack of adaptability of public services that added to the rigidity of its bureaucracy and the closure of some of its general care services. This posed important barriers for IPV care. For future emergency situations, continuity of care should be maintained for these women, access and procedures for processing aid and resources should be facilitated, and there should be better coordination between services, in order to reduce the barriers women face to getting IPV care.

In terms of limitations, it is worth noting that those interviewed were only able to provide a description of events based on the information provided to them by those they had contact with. They were not able to provide information about emerging situations of violence that they did not witness in their workplaces. However, their perceptions are along the lines of what has been observed in other studies of the perceptions of women. In addition, it should also be highlighted that the interviews were carried out between three and six months after the months of COVID-19 confinement. It could be that the perceptions of the problems and facilitators in IPV care were influenced by the post-confinement context. In this study, we aimed to collect the testimonies of a heterogeneous sample that could be considered representative of the diversity of formal IPV services. However, it is necessary to continue to go deeper into the concrete experiences of professionals who came into direct contact with women in situations of social vulnerability due to disability, migratory status, belonging to sexual or ethnic minorities, young women and adolescents and elderly women. Also, we were unable to record the experience of professionals from social and judicial services who play an important role in IPV care. Additional communication with the professionals from the IPV-response services to receive feedback of the findings was not carried out, because of the high workload experienced by the groups during this period.

---

## [Editor Report · Decision Letter 1]

7 Oct 2021

Coping with Intimate Partner Violence and the COVID-19 lockdown: The Perspectives of Service Professionals in Spain

PONE-D-21-22176R1

Dear Dr. Vives-Cases,

We’re pleased to inform you that your manuscript has been judged scientifically suitable for publication and will be formally accepted for publication once it meets all outstanding technical requirements.

Kind regards,

Alok Atreya

Academic Editor

PLOS ONE
---

## [Editor Report · Acceptance letter]

13 Oct 2021

PONE-D-21-22176R1 

Coping with Intimate Partner Violence and the COVID-19 lockdown: The Perspectives of Service Professionals in Spain 

Dear Dr. Vives-Cases:

I'm pleased to inform you that your manuscript has been deemed suitable for publication in PLOS ONE. Congratulations! Your manuscript is now with our production department. 

Kind regards, 

on behalf of

Dr. Alok Atreya 

Academic Editor

PLOS ONE